# The effect of wind direction shear on turbine performance in a wind farm in central Iowa

Miguel Sanchez Gomez[1], Julie K. Lundquist[2,3]

[1]Department of Mechanical Engineering, University of Colorado Boulder, Boulder, 80303, United States
[2]Department of Atmospheric and Oceanic Sciences, University of Colorado Boulder, Boulder, 80303, United States
[3]National Renewable Energy Laboratory, Golden, 80401, United States

*Correspondence to*: Miguel Sanchez Gomez (misa5952@colorado.edu)

**Abstract.** Numerous studies have shown that atmospheric conditions affect wind turbine performance, however, some findings have exposed conflicting results for different locations and diverse analysis methodologies. In this study, we explore how the
change in wind direction with height (direction wind shear), a site-differing factor between conflicting studies, and speed shear affect wind turbine performance. We utilized lidar and turbine data collected from the 2013 Crop Wind Energy eXperiment (CWEX) project between June and September in a wind farm in north-central Iowa. Wind direction and speed shear were found to follow a diurnal cycle, however they evolved differently with increasing wind speeds. Using a combination of speed and direction shear values, we found large direction and small speed shear to result in underperformance. We further analyzed
the effects of wind veering on turbine performance for specific values of speed shear and found detrimental conditions in the order of 10% for wind speed regimes predominantly located in the middle of the power curve. Focusing on a time period of ramping electricity demand (0600 – 0900 LT) exposed the fact that large direction shear occurred during this time and undermined turbine performance by more than 10%. A predominance of clockwise direction shear (wind veering) cases compared to counterclockwise (wind backing) was also observed throughout the campaign. Moreover, large veering was found
to have greater detrimental effects on turbine performance compared to small backing values. This study shows that changes in wind direction with height should be considered when analyzing turbine performance.

## 1 Introduction

Wind power generation directly depends on wind speed. Additionally, power depends on atmospheric conditions like static stability, shear and turbulence (e.g. Bardal et al., 2015; van den Berg, 2008; Kaiser et al., 2007; Rareshide et al., 2009; St.
Martin et al., 2016; Sumner and Masson, 2006; Vanderwende and Lundquist, 2012; Wagner et al., 2010; Walter et al., 2009; Wharton and Lundquist, 2012). Idealized theories state that the power extracted by a wind turbine is a function of the blade element's efficiency (i.e. turbine blade design) and the available power flux through the disk swept by the blades (Burton et al., 2001). However, atmospheric turbine operating conditions diverge from simplified ones used for turbine design. Varying inflow speed and direction profiles, turbulence, transient conditions, and wake effects from upwind turbines alter power
production.

Static stability in the lower planetary boundary layer is governed by temperature gradients that drive or suppress buoyancy (Stull, 1988). Three stability regimes are usually established; stable, neutral, and unstable conditions, corresponding to a stratified, equilibrated, and convective atmosphere, respectively. Several means of quantifying atmospheric stability have been employed in wind energy studies, including the dimensionless wind shear exponent ($\alpha$), turbulence intensity, bulk Richardson number, and the Obukhov length. While some field measurements have provided insight into how stability affects power production, conflicting results have been reported in differing locations. At a wind farm identified only as "West Coast North America", Wharton and Lundquist (2012a,b) found an increase in wind turbine power production during stable atmospheric regimes. In contrast, at another site in the central plains of North America, the opposite effect occurred (Vanderwende and Lundquist, 2012). St. Martin et al. (2016) considered the effects of stability and turbulence at a test site near the Rocky Mountains in Colorado. They found stable conditions to enhance performance near rated speed, while undermining it for lower wind speeds. Their results regarding turbulence effects, however, agreed with theoretical findings by Kaiser et al. (2007) and with observations presented by Rareshide et al. (2009) that suggest a convective atmosphere decreases performance near rated wind speed and causes overperformance near cut-in wind speed.

Several factors could explain the difference in results between these stability regimes. First, the available data and thus the analysis method differs. Wharton and Lundquist (2012b) segregated power production regimes using wind shear exponents and turbulence intensity in a location with channeled flow. In contrast, Vanderwende and Lundquist (2012) employed the Richardson number and wind shear criteria to quantify local atmospheric stability on a wind farm that could experience directional wind shear. The conflicting results among studies suggest that either additional forcing mechanisms are present or that site-specific factors govern the effects on power production. One likely site-specific factor is the role of directional wind shear, which was not explicitly considered in the above studies but could differ between those sites.

Here we seek to resolve these conflicting results by quantifying the role of directional wind shear in turbine performance. Directional shear is the change of wind direction with height. The three main mechanisms for generating direction shear are the thermal wind, the inertial oscillation and surface stress. In the meteorological community, "wind veering" is used to describe the clockwise turning of the geostrophic wind with height, while "wind backing" describes the counter-clockwise turning of the geostrophic wind with height (Holton and Hakim, 2013). Veering tends to be associated with warm air advection, while backing is associated with cold air advection. Both these terms are associated with deep layers in the atmosphere rather than relatively shallow layers of the atmospheric boundary layer (ABL). Within the boundary layer, veering tends to be more common in the Northern Hemisphere due to the direction of the Coriolis force and the resulting Ekman layer.

In a wind energy context, directional shear causes the incoming wind to be misaligned with the rotor axis over some heights of the rotor swept area (Bardal et al., 2015) as the turbine tends to orient itself into the direction of the wind at hub height. Both veering and backing generate a substantial variation of the horizontal wind speed component orthogonal to the turbine axis altering the energy flux through the rotor and the turbine's capability to extract energy (Wagner et al., 2010). Veering decreases the mean relative wind speed experienced by a clockwise-rotating blade, while backing increases it (Wagner et al., 2010). In contrast, the opposite happens for the angle of attack. The angle of attack is larger for veering and smaller for backing

(Wagner et al., 2010). Simulations by Wagner et al. (2010) also show that though backing increases both mean lift and drag over the blade, the resulting tangential force experienced by the rotor is reduced, while it is slightly augmented by veering. The increase in tangential force from wind veering results in a slight increase in power production, whilst wind backing slightly decreases power production.

The existing studies on the effects of atmospheric stability on power production differ in the role of directional wind shear. The Wharton and Lundquist (2012a,b) wind turbines, and St. Martin et al. (2016) testing site were surrounded by complex terrain that steered a channeled flow into the turbines and prevented the development of directional wind shear during stable conditions. In contrast, at the Vanderwende and Lundquist (2012) location, complex terrain did not prevent the occurrence of a changing wind direction with height.

Several methodologies have been employed for studying the effects of directional wind shear on turbine performance. Bardal et al. (2015) used measurements from a test site in the coastline of Norway without distinguishing between veering and backing. They found a small reduced power output below rated speeds for directional shear above 0.05 deg m$^{-1}$. Rareshide et al. (2009) found slight effects on turbine performance for large wind backing values using measurements from several sites across the Great Plains/Midwest region. Walter et al. (2009) characterized directional shear in Texas and Indiana, and used

those findings to run blade-element simulations using the National Renewable Energy Laboratory (NREL) fatigue analysis structures and turbulence (FAST) model. Results from coupling power change simulations with observations evidenced potential power gains as large as 0.5% and losses as low as 3% when considering both speed and direction shear. Simulations by Wagner et al. (2010) employed a simplified model (HAWC2) finding slight increases in power production for veering and larger reductions from backing, for constant wind speed shear values.

In this present study, the effects of directional wind shear on power production were analyzed by separating the effects of speed shear using data collected in the 2013 Crop-Wind Energy eXperiment (CWEX-13) field campaign of a 150 MW onshore wind farm. Section 2 provides an overview of the dataset utilized for this study, which includes turbine power production and wind profiling lidar, and their respective filtering. Section 3 describes the definition of directional wind shear, speed shear and individual turbine's power curves. Wind shear characterization and its effects on turbine power production are summarized in

sections 4 and 5.

## 2 Data

### 2.1 Measurement site

The Crop Wind Energy eXperiment projects (CWEX) in 2010, 2011 and 2013 explored how wind turbines create changes in microclimates over crops (Rajewski et al., 2013, 2014, 2016), how the diurnal cycle affects wind turbine wakes (Lee and

Lundquist, 2017; Rhodes and Lundquist, 2013), and how agricultural cropping and surface management impact wind energy production (Vanderwende and Lundquist, 2016). The 2013 campaign emphasized the impacts of atmospheric conditions like nocturnal low-level jets (Vanderwende et al., 2015) on wind turbine performance and the dynamics of wake variability (Bodini

et al., 2017; Lundquist et al., 2014). These data have also been used to test approaches for coupling mesoscale and large-eddy simulation models (Muñoz-Esparza et al., 2017). The CWEX-13 field campaign took place between late June and early September 2013 in a wind farm in north-central Iowa. Measurements from several surface flux stations, a radiometer, three profiling lidars, and a scanning lidar were collected.

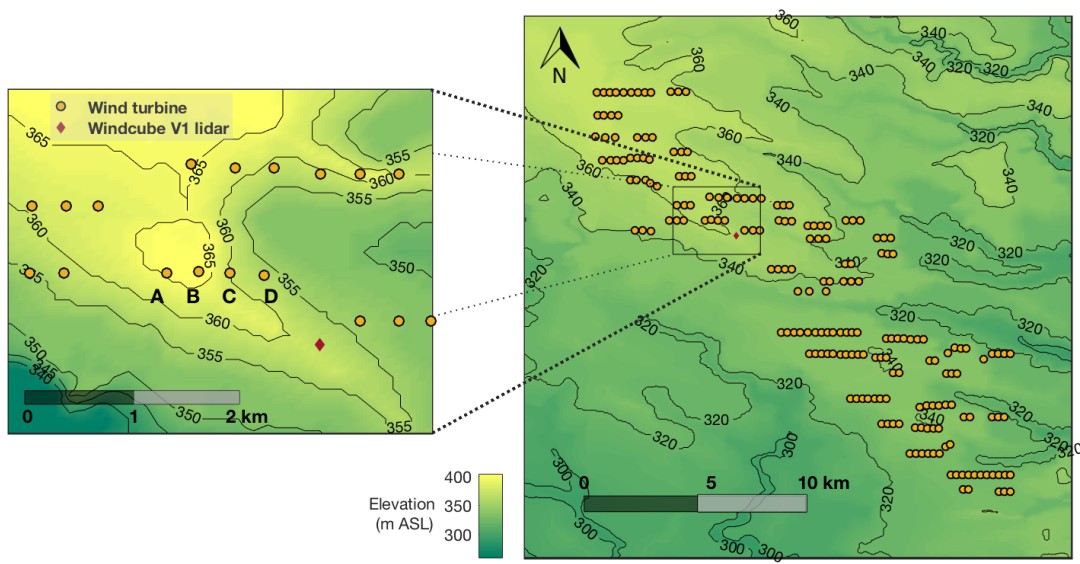

**Figure 1. Schematic view of the wind farm in central Iowa where the CWEX-13 campaign took place. The turbines of interest for this study are marked as A, B, C and D.**

The wind farm consisted of 200 wind turbines extending in a parallelogram with a long axis from the southeast to northwest (Figure 1). The northernmost 100 turbines were General Electric (GE) 1.5 MW extra-long extended (XLE) model and the southernmost 100 turbines were GE 1.5 MW super-long extended (SLE) model turbines. The land was generally flat with a slope smaller than 0.5 deg from southwest to northeast. Turbines were surrounded by a mixture of corn and soybeans, some wetland and lower terrain at the southern edge of the farm, and scattered farmsteads (Rajewski et al., 2013). The region of interest in the 2013 campaign comprised a subset of GE 1.5 MW XLE model turbines (see Table 1 for specifications). For this study, power production from four turbines near one Windcube V1 profiling lidar was utilized.

15 **Table 1. Technical specifications of the turbines studied in the CWEX-13 field campaign (General Electric, 2009).**

| | |
|---|---|
| Rotor diameter (D) | 82.5 m |
| Hub height | 80 m |
| Rated power | 1.5 MW |
| Cut-in wind speed | 3.5 m s$^{-1}$ |
| Rated power at | 11.5 m s$^{-1}$ |
| Cut-out wind speed | 20 m s$^{-1}$ |

## 2.2 Lidar

To quantify wind shear, we relied on data collected from the profiling lidar Windcube V1, designed by Leosphere, deployed during the CWEX-13 campaign. This Doppler wind lidar measured vertical profiles of speed and direction at nominally 1-Hz temporal resolution. It used a Doppler beam swinging (DBS) approach obtaining radial wind measurements along four cardinal

directions at an inclination of 62.5° above the horizon (Vanderwende et al., 2015). The components of the flow were then calculated from the four separate line-of-sight velocities (Lundquist et al., 2015). The CWEX-13 campaign collected wind measurements from 40 to 220 m above ground level at 20-m increments. This study focuses on 2-min average measurements from 40 to 120 m, which comprise the entire turbine rotor layer.

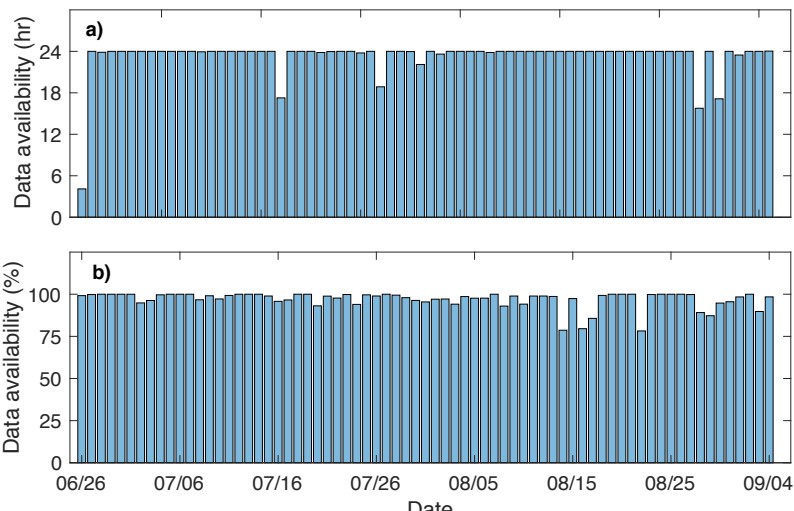

**Figure 2. Lidar (a), and turbine (b) data availability for the duration of the campaign. Turbine power availability only corresponds to percentage of power measurements recorded for wind speeds above 2.5 m s$^{-1}$.**

Wind lidar data was available throughout the campaign (Figure 2a). 2-min wind speed (80 m height) observations for each day only presented significant shortages (>30% of day) for June 27 and August 28. Similarly, power mean data availability for the four analyzed wind turbines and wind speeds above 2.5 m s$^{-1}$ presented great uniformity, having largest shortages

(>20%) for August 14, 16 and 22, and an average 30-min availability of 92%.

The prevailing wind direction for the recorded period in this wind plant was primarily south-southwesterly having a mean wind speed of 8.21 m s$^{-1}$. However, 21% of wind had a strong northerly component (Figure 3), generally associated with frontal passages. The infrequent easterly and westerly winds (90±10 deg and 270±10 deg) were discarded to ensure the turbines were not experiencing wakes from their nearby (within 5 rotor diameters $D$) neighbors. Previous studies in this wind

farm have found wakes in stable conditions to persist for long distances (up to 17.5 $D$) downwind (Bodini et al., 2017), and so therefore all winds with northerly components were also discarded to ensure the profiling lidar was not affected by wakes. Further, only wind speeds between cut-in (3.5 m s$^{-1}$) and cut-out (20 m s$^{-1}$) were considered.

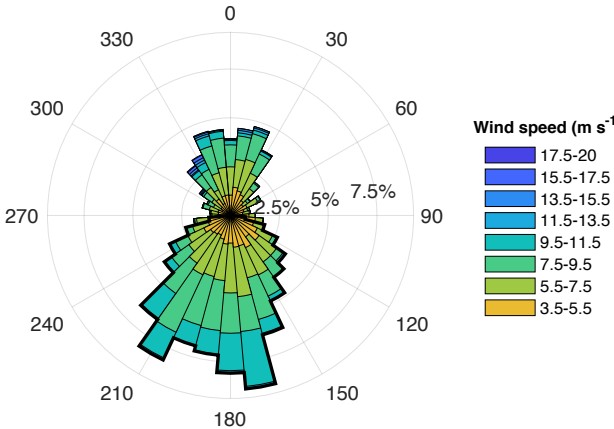

**Figure 3. Wind rose for lidar hub-height altitude (80-m) measured wind speeds between cut-in and cut-out. The black outline highlights the wind direction sector (southerly) used for subsequent data analysis.**

The remaining analysis only considers winds with southerly components (wind direction between 100 and 260 deg), and speeds between 3.5 and 20 m s$^{-1}$. Of note, most winds above 11 m s$^{-1}$ occurred with the northerly frontal passages, so this direction filtering also effectively restricts the analysis to wind speeds below 11 m s$^{-1}$.

## 2.3 Wind turbines

The subset of turbines employed for this study consists of four clockwise-rotating (while looking downwind) GE XLE 1.5-MW, variable blade pitch wind turbines (see Table 1 for specifications). Power production, nacelle wind speed and blade pitch angles were provided by the wind farm operator as 10-min averages recorded via the supervisory control and data acquisition (SCADA) system of each turbine. To analyze how wind shear impacts power production, turbine underperformance during curtailments was filtered following the blade pitch angle approach of St. Martin et al. (2016). Blade pitch angles are controlled to maximize power production as a function of nacelle-measured wind speed, and large blade pitch angles typically represent curtailed conditions or rapidly changing conditions. Therefore, we discarded 10-min periods with blade pitch angles outside $\pm 4.5$ the mean absolute deviation (MAD) for each 0.5 m s$^{-1}$ wind speed bins (Figure 4).

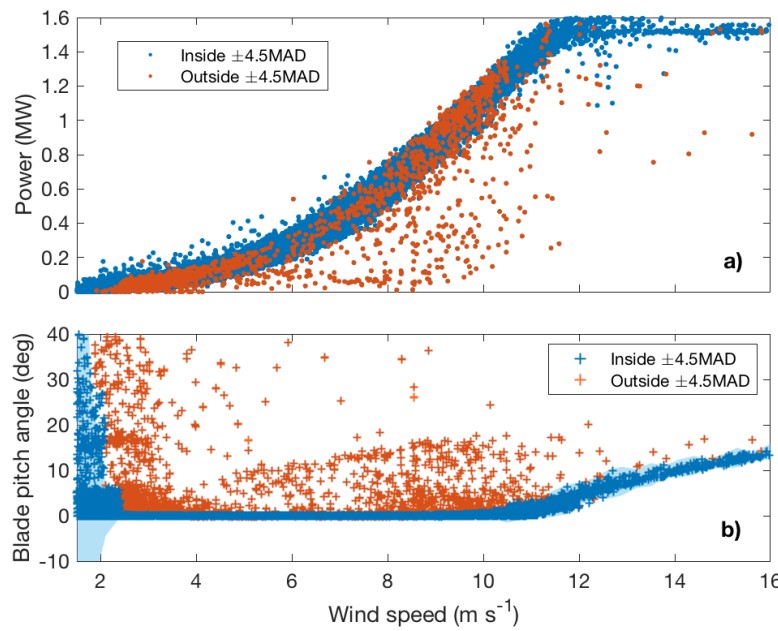

**Figure 4. Power curve based on nacelle-measured wind speed (a), and blade pitch angle from a single blade (b) combining data from the four analyzed wind turbines. The blue envelope in the bottom graph represents $\pm4.5$ MAD of the blade pitch angle within 0.5 m s$^{-1}$ wind speed bins. Red scatter points show 10-min periods filtered out for curtailments, represented by data points outside the MAD envelope.**

## 2.4 Time averaging

Turbine- and lidar-recorded data are averaged over different time intervals by their respective data-acquisition systems (2-min for lidar, and 10-min for turbine). Matching turbine performance with atmospheric conditions was performed by averaging 2-min lidar measurements for the corresponding 10-min turbine power production period. For example, turbine data for July 04, 2013 from 0500 to 0510 LT is matched with the average of five 2-min lidar data measurements corresponding to the same date and time period. As is illustrated in Figure 5, turbine and lidar data were synchronized for the duration of the campaign.

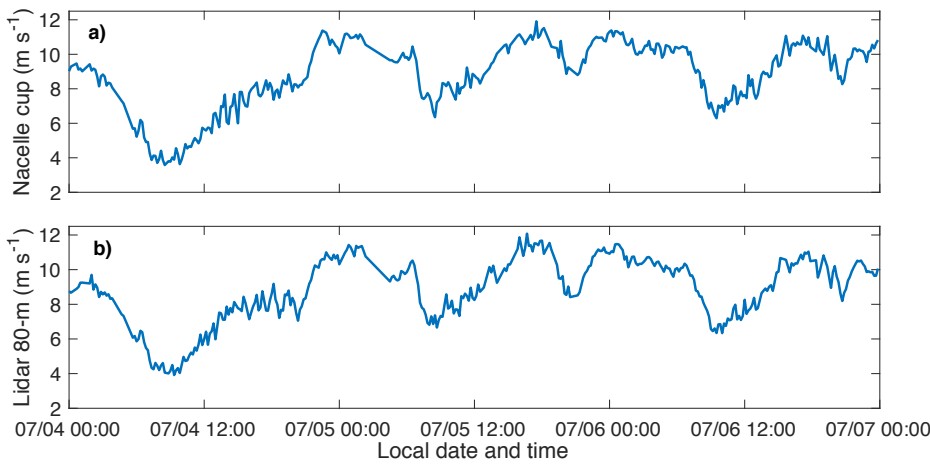

**Figure 5. Time series from 00:00 July 04, 2013 to 00:00 July 07, 2013 of hub-height wind speeds measured by the cup anemometer on the nacelle of Turbine A (a), and 80-m wind speeds measured by the lidar (b).**

## 3 Methods

### 3.1 Turbine power curves

According to the International Electrotechnical Commission's Wind Turbine Power Performance Standard (2005), wind turbine power performance characteristics are determined both by the measured power curve and the annual energy production. The measured power curve is obtained by simultaneously collecting data from meteorological variables and turbine performance over long periods of time. Wind speed is measured at hub height using cup anemometers mounted on a meteorological mast positioned 2 – 4 rotor diameters upwind of the turbine, and power output is recorded using a power measurement device (e.g. power transducer) between the wind turbine and the electrical connection. Measurements are averaged over 10-min time periods. A database for a wide range of wind speeds (0.5 m s$^{-1}$ bins) is used to establish the relationship between the nacelle-height wind speed and wind turbine power output. General Electric's power curve for the 1.5 MW wind turbines in this study is shown in Figure 6 as the dashed line.

Power production for the duration of this campaign reflected persistent differences from the manufacturer's reference values at wind speeds below 8 m s$^{-1}$ and above rated (e.g. Turbine C in Figure 6). Consequently, mean power curves for each turbine were utilized as a reference value for normalization to have a consistent comparison of performance among individual devices. Turbine power curves using nacelle-measured wind speed, and lidar-measured wind speed filtering easterly, westerly and northerly winds were compared. The Pearson correlation coefficient was used in each case to determine which power curve showed highest correspondence to the manufacturer's power curve for wind speeds below rated. Power curves obtained using lidar-measured wind speed displayed higher resemblance to GE curves (average $\rho = 0.9590$, p-value = 0.00) than

power curves obtained from nacelle-measured wind speed (average $\rho = 0.9061$, p-value = 0.00). Therefore, normalization was performed with respect to each turbine mean power curve obtained using lidar-measured wind speed.

As suggested by the histogram in Figure 6, the frequency of occurrence changed with wind speed, roughly following a Weibull distribution with a shape factor of 2.05 and a 6.9 m s$^{-1}$ scale parameter. To determine if the sample size (number of power observations) for each 0.5 m s$^{-1}$ wind speed bin was large enough for estimating each turbine's population mean (observed power curve), we calculated the required sample size to have a 99.5% confidence that the error ($e$) in the observed mean power does not exceed half the difference of mean power between two adjacent wind speeds. The power estimator ($\overline{p}$) is assumed to be a normally distributed estimator of the real turbine power ($p$) for each wind speed bin, then their difference is a normal distribution (Walpole, 2007):

$$\frac{\overline{p}-p}{\sqrt{Var(\overline{p})}} \sim N(0,1) \, . \tag{1}$$

The allowable error was designated as half the difference in mean power between two adjacent wind speeds $\left(e_V = 0.5\left(\overline{p}_{V+0.5} - \overline{p}_V\right)\right)$, and so the probability of the real and observed mean difference being greater than the allowable error $\left(P\left(|\overline{p}_V - p_V| > e_V\right)\right)$ is 0.005 (i.e. 99.5% confidence). A property of a normal distribution is that this same probability holds for $|\overline{p}_V - p_V| > z_{\alpha/2}\,\sigma_V/\sqrt{n_V}$. Thus, the minimum sample size to have 99.5% confidence that the error in the observed mean power does not exceed half the difference of mean power between two adjacent wind speeds is $n_V = z_{\alpha/2}\,\sigma_V\sqrt{e_V}$. Every turbine had sufficient data points for wind speeds between 4 and 11 m s$^{-1}$ (referred to as the partial load regime), and every 0.5 m s$^{-1}$ bin within this range had at least 106 observations, supporting the assumption for a normally distributed estimator. Turbine power output for wind speeds outside this range (e.g. turbine overperformance between 12 and 15 m s$^{-1}$ in Figure 6) is not well represented by our data given that there were insufficient observations to derive a power estimator that meets the aforementioned criteria. The remaining analysis only considers the region of the power curve where the observed mean power accurately represents the real mean turbine power, effectively restricting the subsequent analysis to winds below rated speed.

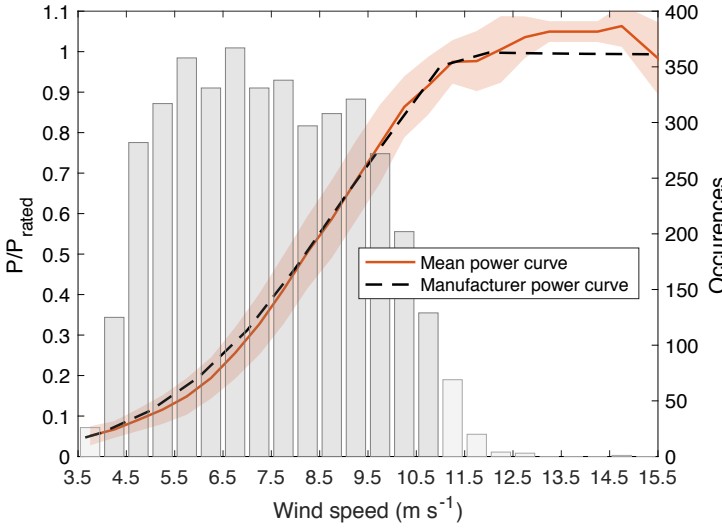

**Figure 6. Mean power curve for Turbine C based on 80-m lidar wind speed measurements overlaid over the number of power production cases for each 0.5 m s⁻¹ wind speed bins. The shaded region in the power curve corresponds to ±1 standard deviation, and the dark grey histogram corresponds to the wind speed ranges considered in this study.**

## 3.2 Wind shear

Directional wind shear is defined as the change in wind direction with height, and speed shear corresponds to the change in the mean horizontal wind speed. One mechanism for generating wind shear is the vertical shear of geostrophic wind referred to as thermal wind. The thermal wind is caused by large-scale horizontal temperature gradients that can be created by sloping terrain, fronts, land-sea interfaces, and large-weather patterns (Stull, 1988). Wind shear overnight is also generated by the inertial oscillation (Blackadar, 1957; Van de Wiel et al., 2010). The inertial oscillation is the rotation in the wind vector in the residual layer caused by a force imbalance at sunset, when mixed layer turbulence ceases. As frictional stress diminishes after sunset, pressure gradients tend to accelerate subgeostrophic winds in the mixed layer back toward geostrophic. Inertia from the counteracting Coriolis force induces an oscillation in the wind vector causing it to become supergeostrophic and to turn clockwise (northern hemisphere) with time (Stull, 1988). A third forcing mechanism is frictional drag with the ground. Turbulent momentum fluxes in the boundary layer reduce the actual wind speed near the surface. The Coriolis force, being directly proportional to the wind speed, decreases creating a force imbalance with the pressure gradients. As a result, the actual surface wind vector is directed across the isobars toward low pressure (Holton and Hakim, 2013).

Directional shear in this study is calculated as the shortest rotational path between wind vectors at 40 and 120 m above ground level, normalized over vertical distance between the measurements. For example, a case with southerly winds at 40 m and westerly winds at 120 m would be calculated as 90 deg shear over the 80 m layer depth, or 1.125 deg m⁻¹. We characterize speed shear through the dimensionless wind shear exponent $\alpha$ using the power law expression,

$$V = V_R \left(\frac{z}{z_R}\right)^{\alpha}, \tag{2}$$

where $V$ is the mean horizontal wind speed at height $z = 120$ m, and $V_R$ is the mean horizontal wind speed at reference height $z_R = 40$ m above ground level.

Speed and direction wind shear alter the available power of the air through the turbine and its ability to extract energy from the wind (Wagner et al., 2010). The available power in the air flowing across a disc is proportional to the projection of the velocity vector over the disc area,

$$p_{avail.} \propto \left( \vec{V} \cdot \vec{n} \right)^3, \tag{3}$$

where $\vec{V}$ is the wind vector, $\vec{n}$ is the unit vector normal to the disc area, and $p_{avail.}$ is the available power in the air. Several models have shown that speed shear exponents above 0 and below 0.33 result in lower available power over the whole rotor area, whereas larger $\alpha$ values increase the energy flux compared to a uniform flow with hub-height speed (e.g. Antoniou et al., 2009; Bardal et al., 2015). Blade aerodynamic performance with shear also diverges from design conditions. Changing wind direction and speed with height makes the relative velocity between the air and the blades, and the effective angle of attack to vary (Wagner et al., 2010), causing the turbine blades operate at suboptimal blade pitch angles.

The literature includes a range of different classification thresholds to analyze contrast high wind shear and low wind shear to explore its effects on turbine performance. Bardal et al. (2015) utilized a threshold of 5 deg over a vertical extent of 100.6 m (or 0.0497 deg m$^{-1}$) to distinguish between high and low direction shear scenarios in a wind farm on the coastline of mid-Norway. They found small detrimental effects of high veering on power production for wind speeds near 7, 8 and 9.5 m s$^{-1}$ (Bardal et al., 2015). To examine the effects of speed shear, they considered different ranges of the power law exponent and found a reduction of turbine efficiency for high-shear ($\alpha > 0.15$) conditions in the partial load regime (Bardal et al., 2015). Rareshide et al. (2009) considered a statistical description specific to several sites across the Great Plains/Midwest region, encountering slight performance reductions for wind backing of -0.25 deg m$^{-1}$ and near uniform speed profiles. Walter et al. (2009) performed blade-element modeling using the fatigue analysis structures and turbulence model (FAST) from the National Renewable Energy Laboratory to quantify the effects of speed and direction shear on performance. Simulation results for 8 and 10 m s$^{-1}$ wind speeds showed a maximum instantaneous 6% underperformance occurring for wind speed shear exponents of 0.35 and wind backing of -0.472 deg m$^{-1}$. Here, we considered the combined effect of direction and speed shear on turbine performance to define the shear values that segregate under- and overperformance in this wind farm.

## 4 Results

### 4.1 Wind shear characterization

A predominance of wind veering was observed in this site compared to wind backing cases (Figure 7a). Wind veering occurred more than 77% of the time and displayed larger numerical mean and maximum values (0.0939 deg m$^{-1}$ and 1.83 deg m$^{-1}$, respectively) compared to backing (-0.0144 deg m$^{-1}$ and -1.17 deg m$^{-1}$, respectively).

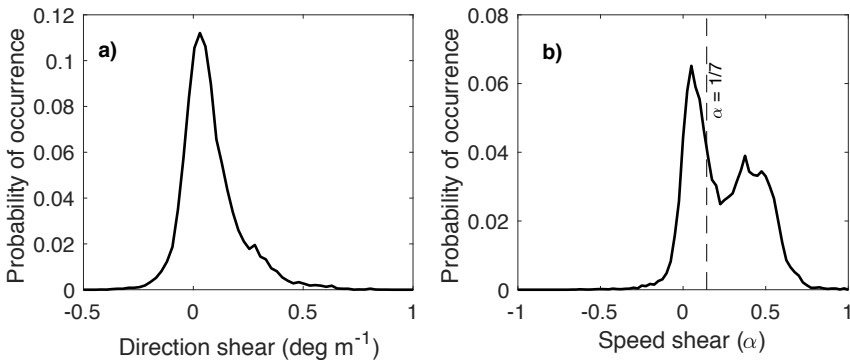

**Figure 7. Probability distribution for direction shear (a), and speed shear (b) in the rotor layer (40 – 120 m above ground level).**

The speed shear probability distribution was bimodal, with a narrow peak centered around 0 and a broad peak close to 0.4 (Figure 7b). An increase of speed with height was observed 88.6% of the time, from which 53% was above 0.225. Further, 60% of the recorded data lays above the commonly used 1/7 power law exponent.

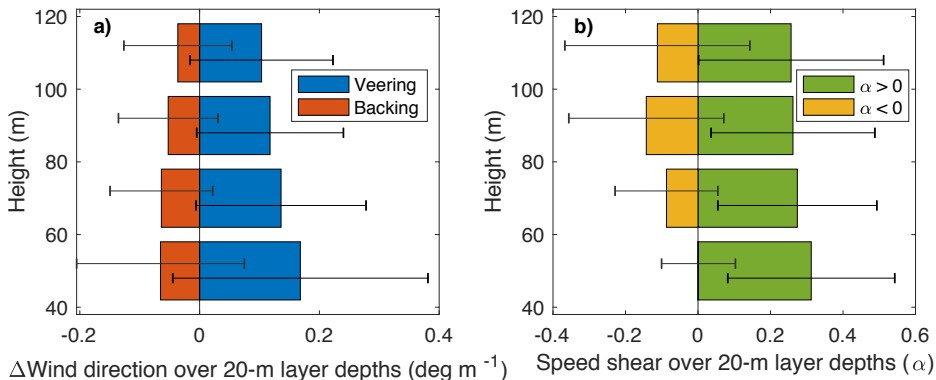

**Figure 8. Wind direction (a), and speed shear (b) evolution with height across the rotor layer.**

Both direction and speed shear had a tendency to decrease with height. Figure 8a illustrates how wind direction evolved differently through the rotor layer for veering and backing cases. Both clockwise and counterclockwise wind direction rate of change were larger in the lower rotor layer. Directional shear was 1.6 times larger from 40 to 60 m compared to 100 to 120 m above ground level for veering, and 1.79 times larger for backing. When considering the absolute value of the wind vector rotation, the lower layer (40 to 60 m) experienced an average change in wind direction 1.55 times larger than the upper layer (100 to 120 m). Figure 8b demonstrates how wind speed changed unevenly for positive and negative power law exponents. Negative power law exponent cases only started evidencing decreasing wind speeds with height above 60 m, whereas positive $\alpha$ values presented the largest rate of change in the lower rotor layer. Speed shear was 1.2 times larger from 40 to 60 m compared to 100 to 120 m above ground level for positive shear values.

Direction and speed shear at the test site varied accordingly with time of day (Figure 9). The correlation between both parameters is 0.9. Nighttime cases showed an evolving surface layer that does not reach equilibrium, as is depicted by consistently increasing directional shear across the rotor layer at an average rate of 0.0304 deg m$^{-1}$ hr$^{-1}$, and 0.0117 hr$^{-1}$ for speed shear from before sunset until just after sunrise. Daytime cases, on the other hand, experienced a rapid morning transition following sunrise (-0.1171 deg m$^{-1}$ of directional shear, and -0.0723 of speed shear per hour) followed by a fairly consistent surface layer having a slowly decreasing mean directional shear of 0.004 deg m$^{-1}$ per hour (0.0082 increase of $\alpha$ per hour). Average changes in wind direction (speed) with height at night was 2.6 (4.2) times larger than during daytime after the morning transition period.

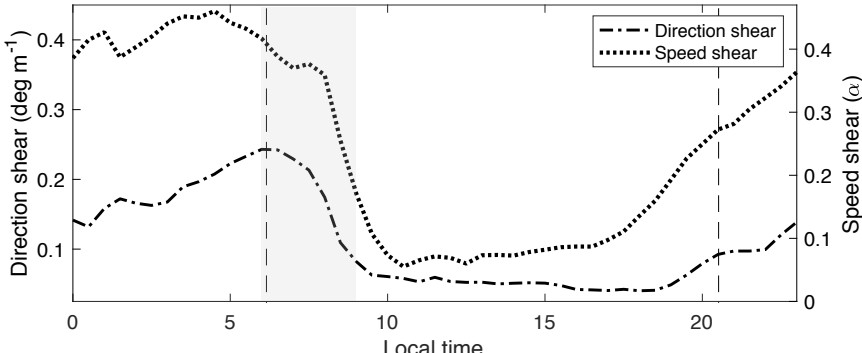

**Figure 9. Diurnal cycle of mean wind direction and speed shear for wind speeds between cut-in and cut-out. Dashed vertical lines indicate sunrise and sunset times for August 1, 2013, the mid-point of the dataset analyzed here. The grey shaded region indicates the morning transition period (0600 – 0900 LT).**

Of particular interest is the morning time period (from 0600 to 0900 local time) which, according to the U.S. Energy Information Administration (2019), experiences increasing electricity demand in the Midwest region. Wind shear presented its largest rate of change during this time period (Figure 9). At this time, nearly 50% of the recorded data between cut-in and cut-out wind speed was within 5 and 8 m s$^{-1}$, with a mean direction and speed shear of 0.196 deg m$^{-1}$ and 0.37, respectively. This average shear exceeded the mean daytime (0.0838 deg m$^{-1}$, 0.168) and whole-day (0.1137 deg m$^{-1}$, 0.258) values, and nighttime (0.1613 deg m$^{-1}$, 0.39) direction shear value.

Though speed and direction shear varied proportionally throughout the day, they had opposite monotonical relationships with wind speed. As wind speed increased, so did speed shear, but direction shear decreased (Figure 10). Directional shear declined with increasing wind speed for both daytime and nighttime cases. While directional shear at night was generally larger than during the day, in both cases direction shear decreased at a median rate of around 0.0166 deg m$^{-1}$ for each increase in m s$^{-1}$ in wind speed. The power law exponent increased proportionally with wind speed at a rate of 0.0672 during nighttime for speeds below 9 m s$^{-1}$, and then stabilized. During daytime, a growth of 0.0184 in $\alpha$ occurred for each increase in m s$^{-1}$ in wind speed up to 7.5 m s$^{-1}$; for higher speeds, speed shear decreased at a rate of -0.0195 for each increase in m s$^{-1}$. Daytime is defined as the period between sunrise as sunset for each date, and nighttime corresponds to the complementary period. Daily sunrise and sunset information were estimated using NOAA's sunrise/sunset calculator (National Oceanic and Atmospheric

Administration, 2019). Median values appear in Figure 10 rather than mean values as the data presented a large spread with a large percentage of outliers. Outliers are considered observations outside the quantile 3 (75th percentile range) plus or minus a predetermined interquartile range (range between 25th and 75th percentile) for each 0.5 m s$^{-1}$ wind speed bins (Q3±1.5IQ).

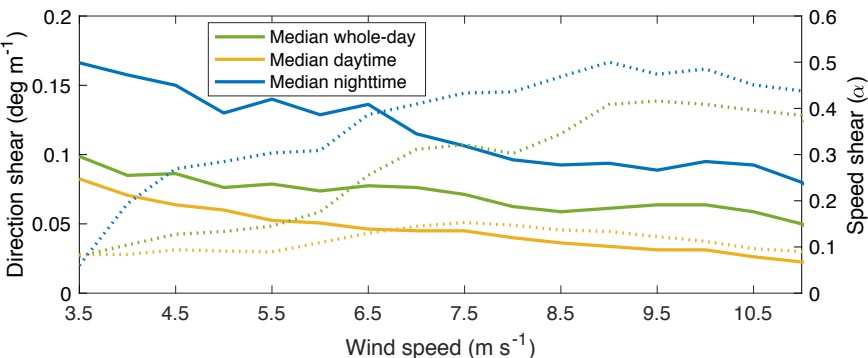

5    **Figure 10. Direction (solid line) and speed (dotted line) wind shear variation with 80-m wind speed using each day's sunrise/sunset times of day.**

Nighttime shear exceeded that during the day for wind speeds between cut-in and rated speed (Figure 10). Median nighttime directional wind shear was at least 1.8 times as large as daytime cases for wind speeds between cut-in and rated speed. The highest percentage difference occurred near rated wind speeds, where nighttime directional shear was 3.5 times larger than

10    that during the day. Median speed shear during the night was on average 3.2 times larger than during daytime and presented the largest differences near rated speeds (about 4 times larger for wind speeds between 8 and 11 m s$^{-1}$).

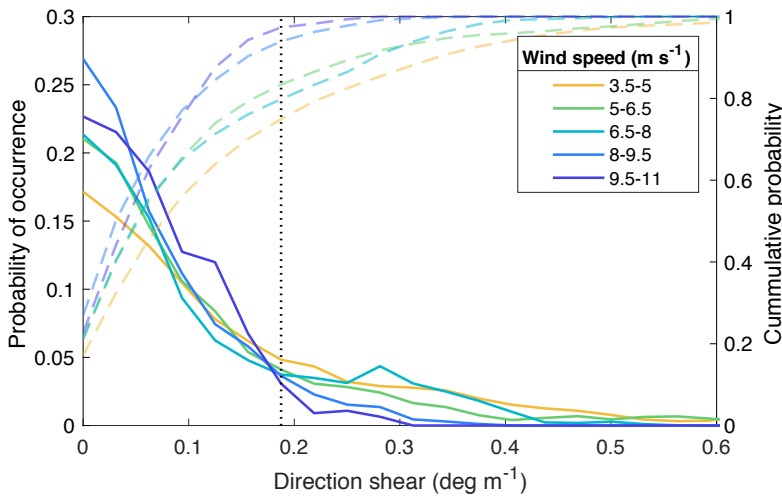

**Figure 11. Directional wind shear probability density (solid lines) and cumulative (dashed lines) distributions for 1.5 m s$^{-1}$ wind speed regimes. The black dotted line marks 0.1875 deg m$^{-1}$ of directional wind shear.**

15    Large directional wind shear tended to occur at wind speeds below 8 m s$^{-1}$ (Figure 11). The number of occurrences of directional wind shear cases above 0.1875 deg m$^{-1}$ in this site varied considerably for wind speeds above and below 8 m s$^{-1}$.

The number of observed cases of directional shear larger than 0.1875 deg m[-1] followed a similar trend for wind speeds between cut-in and 8 m s[-1], accounting for approximately 20% of observations. Conversely, wind speeds between 8 m s[-1] and rated speed reported considerably fewer cases above 0.1875 deg m[-1] of directional shear (∼4% of observations for each 1.5 m s[-1] wind speed bin).

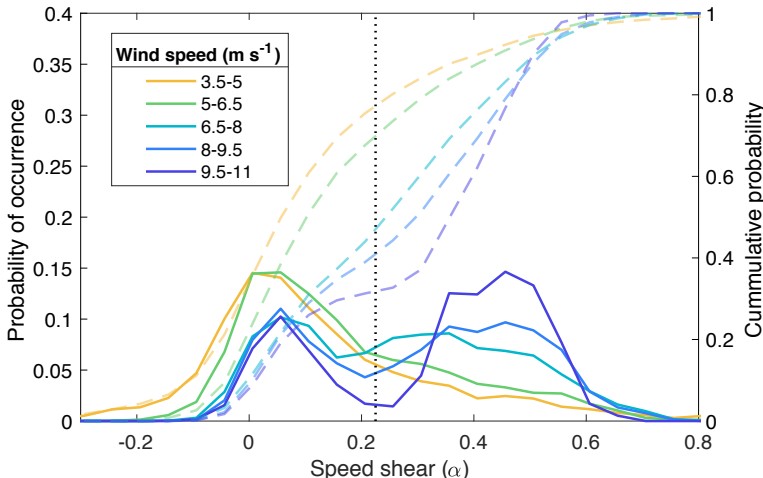

**Figure 12. Speed shear probability density (solid lines) and cummulative (dashed lines) distributions for 1.5 m s[-1] wind speed regimes. The black dotted line marks $\alpha$ = 0.225.**

Speed shear distributions changed dramatically for wind speeds above and below 6.5 m s[-1] (Figure 12). Wind speeds near cut-in depicted a single peak centered at zero-shear with a broad right tail. Above 6.5 m s[-1] wind speeds, power law exponent

10    density distributions were bimodal, where increasing speeds displayed a trend for more cases of large shear values. Cumulative probability distributions for moderate and large wind speeds (> 6.5 m s[-1]) displayed a curvature change from concave to convex around $\alpha$ = 0.225. More than 70% of observations occurred at speed shear values below 0.225 for wind speeds between cut-in and 6.5 m s[-1]. In contrast, only 32% of the recorded data presented shear values below 0.225 for wind speeds above 9.5 m s[-1], and less than 50% for 6.5 – 9.5 m s[-1] wind speeds.

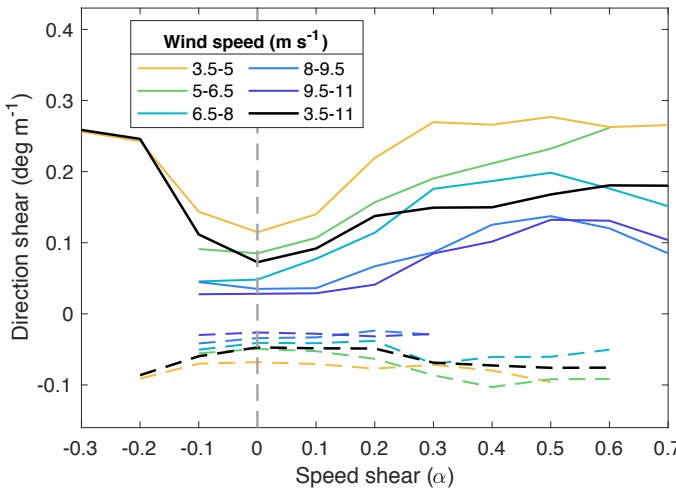

**Figure 13. Mean speed and direction shear relationship for similar hub-height wind speed regimes (1.5 m s⁻¹ bins). Solid lines correspond to wind veering and dashed lines to wind backing. Black lines correspond to the mean for all wind speeds. The dashed grey line marks zero-speed shear value.**

Both shear parameters were correlated for similar hub-height wind speed regimes (Figure 13). The correlation coefficient for increasing speed shear and direction shear values (veering and backing) is 0.9, and for decreasing direction shear and increasing speed shear (veering and backing) is -0.9. The largest rate of change of shear parameters occurred for negative $\alpha$ values, where direction shear increased at an average rate of 0.735 deg m⁻¹ (0.607 deg m⁻¹) per unit decrease (increase) of speed shear for veering (backing). Positive power law exponents displayed a small mean rate of change per unit increase of

speed shear for veering and backing (0.165 deg m⁻¹ and 0.063 deg m⁻¹, respectively). Further, smaller wind speeds evidenced a stronger relationship between speed and direction shear. Mean veering increased at an average rate of 0.347 deg m⁻¹ per unit increase of speed shear for wind speeds below 8 m s⁻¹ ($0 \leq \alpha \leq 0.4$), whereas veering increased at a rate of 0.21 deg m⁻¹ for speeds above 8 m s⁻¹. Mean backing displayed an additional dissimilarity for $0 \leq \alpha \leq 0.4$, where a positive correlation existed near-rated speeds (>8 m s⁻¹), and a negative one for lower wind speeds (0.04189 deg m⁻¹ and -0.0709 deg m⁻¹ per unit increase

in $\alpha$, respectively).

## 4.2 Effects on turbine performance

    We normalized each power measurement to quantify the role of wind direction shear and speed shear on turbine power production for different wind speeds. Normalized performance ($\hat{p}$) is defined as the ratio of each 10-minute power observation ($p$) and the mean power estimator ($\overline{p}$) of the corresponding 0.5-m s⁻¹ wind speed bin at which it occurred (Eq. (4)). With this

approach, underperformance for a given wind speed is defined as turbine power production smaller than the mean ($\hat{p} < 1$), and overperformance as turbine power production larger than the mean ($\hat{p} > 1$).

$$\hat{p}(v_i) = \frac{p(v_i)}{\overline{p}(v_i)}. \tag{4}$$

Segregating normalized turbine power into speed shear ($\alpha$) and direction shear ($\beta$) combinations revealed a threshold (referred to as $\alpha/\beta$ threshold from now on) that separates over- and underperformance at this wind farm (Figure 14). Speed and direction shear combinations that satisfy Eq. (5) tended to result in turbine performance equal to the mean observed throughout the campaign:

$$\beta = \frac{2}{3}\alpha - 0.1 \ . \qquad\qquad\qquad\qquad\qquad\qquad\qquad\qquad\qquad (5)$$

Turbine performance for atmospheric conditions that lay above the $\alpha/\beta$ threshold in Figure 14 resulted in underperformance for this dataset. Mean normalized power above and below the threshold was 0.94 and 1.01, respectively. Furthermore, this threshold allowed to distinguish power production in a statistically significant way (99.99% significance) for above- and below-threshold cases. A multiway analysis of variance revealed that both speed and direction shear affect the

mean of normalized turbine performance (turbines A, B, C and D) for observations above and below the $\alpha/\beta$ threshold. Individual turbines' normalized performance evidenced similar results as combining the altogether (not shown). Every analyzed turbine displayed significant differences for normalized turbine power for cases above and below the threshold (99.99% significance).

Small wind backing and small veering showed similar effects on turbine performance (Figure 14). Veering below 0.1 deg

m$^{-1}$ and backing above -0.1 deg m$^{-1}$ only reported statistically distinct (1% significance) normalized performance for speed shear exponents between 0.3 – 0.4 and 0.5 – 0.6. Clockwise direction shear resulted in slight overperformance (1.00 and 1.06 for $0.3 < \alpha < 0.4$ and $0.5 < \alpha < 0.6$, respectively). In contrast, counterclockwise direction shear resulted in underperformance (0.90 and 0.97 for $0.3 < \alpha < 0.4$ and $0.5 < \alpha < 0.6$, respectively). Mean negative direction shear was near-zero for both speed shear ranges (-0.036 deg m$^{-1}$ for $0.3 < \alpha < 0.4$, and -0.035 deg m$^{-1}$ for $0.5 < \alpha < 0.6$). Further, mean normalized turbine power

production was 0.9806 for all backing observations throughout the campaign. Veering observations displayed similar results as mean normalized performance was 0.9877. Atmospheric conditions for all wind veering observations were predominantly just above the $\alpha/\beta$ threshold. Mean direction and speed wind shear were 0.118 deg m$^{-1}$ and $\alpha = 0.31$, respectively. Wind backing cases were generally below the $\alpha/\beta$ threshold ($\alpha = 0.12; \beta = -0.031 \deg \mathrm{m}^{-1}$).

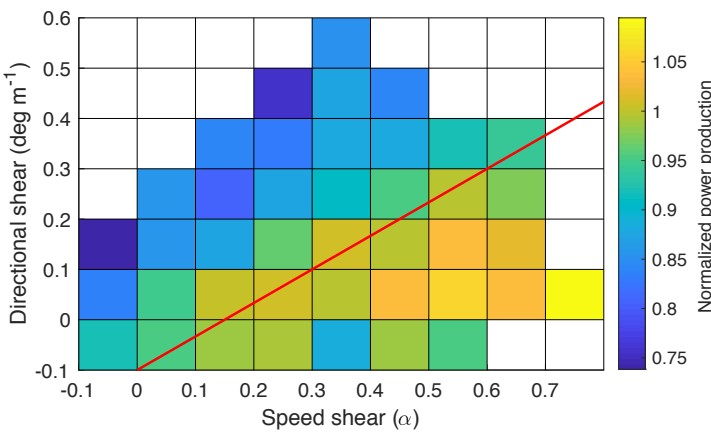

**Figure 14. Mean normalized power production (turbines A, B, C and D) for all combinations of speed and direction shear that present more than 30 observations. The red line represents the $\alpha/\beta$ threshold.**

Because one of the main differences between the Wharton and Lundquist (2012b) and Vanderwende and Lundquist (2012) studies was the occurrence of directional wind shear at the different sites, we here examined the effect of the shear of wind direction on turbine performance. To separate the effect of speed shear from that of direction shear, we isolated turbine performance that transpired within a 0.1 power law exponent interval and segregated observations using the $\alpha/\beta$ threshold for each speed shear bin. Figure 15 illustrates how turbine performance was undermined with larger directional wind shear for different wind speed regimes. Mean normalized power was statistically distinct (1% significance) for wind speeds between 5.5 – 6 m s$^{-1}$ and 7.5 – 8 m s$^{-1}$, and speed shear ranges (0.1 bins) between 0.2 and 0.4. Moreover, turbine performance differed (99% confidence) for 5.5 – 9 m s$^{-1}$ wind speed regimes when considering power law exponents between 0.2 and 0.3 (Figure 15a). Normalized mean turbine performance for these wind speed regimes was 1.03 and 0.85 for cases below and above the threshold, respectively. Larger speed shear (Figure 15b) presented additional differences for smaller and larger wind speeds (4 – 4.5 m s$^{-1}$, 9 – 10 m s$^{-1}$ and 11 – 11.5 m s$^{-1}$), however speeds in the middle of the partial load regime did not present as many significant differences (1% significance). Normalized mean turbine performance was 1.02 and 0.90 for statistically distinct wind speed regimes below and above the threshold, respectively.

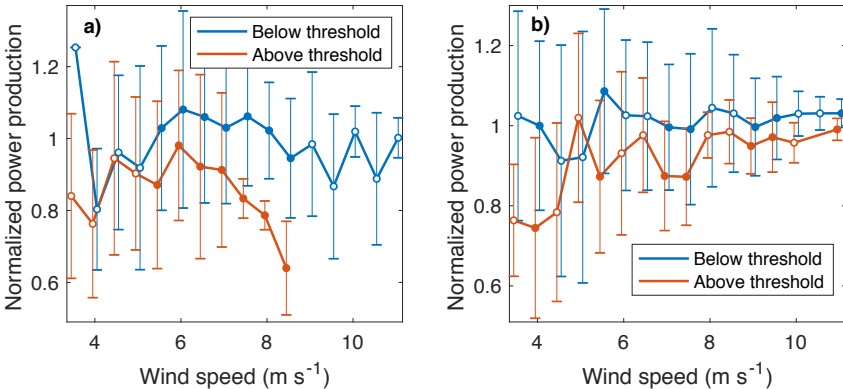

**Figure 15. Mean normalized power production (turbines A, B, C and D) of observations above and below the $\alpha/\beta$ threshold for speed shear between 0.2 – 0.3 (a), and 0.4 – 0.5 (b). Filled circles are statistically distinct. Errorbars represent 1 standard deviation from the mean.**

Power variability can exert significant impact during morning hours, when power demand tends to increase. Throughout all power production periods, from 0600 to 0900 local time, 47.6% of the dataset reported speed and direction shear combinations above the $\alpha/\beta$ threshold. Further, normalized performance for $0.3 < \alpha < 0.4$, the power law exponent bin that presented the greatest number of observations above the threshold just after sunrise, depicted statistically distinct (1% significance) values for wind speed regimes between 5 – 6.5 m s$^{-1}$, 7.5 – 9 m s$^{-1}$ and 10 – 10.5 m s$^{-1}$ for above- and below-threshold cases (Figure 16). Normalized mean turbine performance for these wind speed regimes was 1.08 and 0.87 for cases below and above the threshold, respectively.

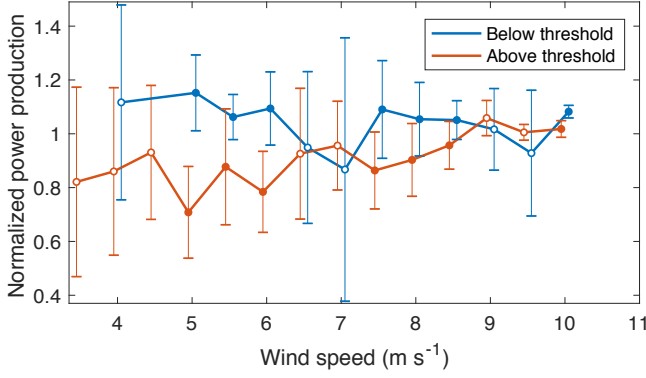

**Figure 16. Mean normalized power production (turbines A, B, C and D) of observations above and below the $\alpha/\beta$ threshold for speed shear between 0.3 – 0.4 during the morning transition (0600 – 0900 LT). Filled circles are statistically distinct. Errorbars represent one standard deviation from the mean.**

## 5 Discussion and Conclusions

Wind shear at the test site showed more veering cases than backing cases (Figure 7a) and a predominance of wind speed increasing with height (Figure 7b), as would be expected from the balance between Coriolis, pressure gradient, and frictional forces in the atmospheric boundary layer (Holton and Hakim, 2013). Furthermore, the largest shear values occurred between 40 m and 60 m above ground level (Figure 8), as would also be expected given that turbulent fluxes increase near the surface, causing larger wind vector rotation and speed reduction. However, cases where wind speed decreased between 40 m and 120 m evidenced the greatest rate of change between 80 m and 100 m above the surface (Figure 8b). These observations usually took place at low wind speeds during the middle of the day, where a highly convective boundary layer produces near-zero shear in the lower rotor layer.

Shear also depended on time of day (Figure 9). The observed diurnal pattern is consistent with daily radiative flux cycles. The advent of shortwave radiation from the sun at dawn drives convective air plumes from surface heating causing the largest rate of shear decrease. Rising air parcels transport air with similar zonal and meridional speed components across the rotor layer, decreasing wind shear. As the sun continues to heat the surface through the day, the convective atmosphere is strengthened, and wind direction and speed shear tend to stabilize. Once the short-wave radiative flux ceases at dusk, atmospheric stratification develops, evident from increasing shear values. A previous study in this same site found stable stratification to develop at 1900 local time, and strong veering and speed shear to develop after the evening transition (Lee and Lundquist, 2017). Median nighttime direction and speed (above 4 m s$^{-1}$) shear were at least 1.8 and 2.3 times larger, respectively, than daytime (Figure 10), consistent with decoupled surface and residual layers within the atmospheric boundary layer. The upper portion of the ABL starts decoupling from that close to the ground as convective turbulent fluxes no longer maintain homogeneity in the atmosphere. Vanderwende et al. (2015) found strong, persistent low-level jets during nighttime at this site, which tend to further increase shear compared to daytime cases. Changes in wind direction and speed with height tended to increase throughout the night, suggesting that the rotor-layer never equilibrated during nighttime.

Though speed and direction shear in the boundary layer have equivalent forcing mechanisms, they displayed opposite relationships with increasing wind speeds (Figure 10). Convective conditions, typically with low speed shear, usually occurred at low wind speeds (Figure 12), where large direction shear was more likely (Figure 11). Large convective eddies cause a fluctuation of the meridional and zonal speed components (large direction shear), but mean horizontal wind speeds remain almost unchanged (small speed shear). Figure 12 suggests that a stratified layer, which entails large wind speed shear, primarily occurred near rated speeds. Decoupled laminar flow through the rotor layer results in low direction shear, whereas winds accelerate toward supergeostrophic speeds (large speed shear).

Nevertheless, a monotonic relationship between speed and direction shear existed for similar hub-height wind speed regimes (Figure 13). As wind profiles evolved for constant hub-height speeds, both shear parameters developed congruently due to the force balance in the boundary layer. Large surface stress reduces wind speed near the ground and results in cross-isobaric flow toward low pressure. At higher altitudes above ground level, the surface stress is lower, and the wind is

geostrophic. In between these heights, the variation of speed and direction with height is described by the Ekman spiral, where wind vectors must increase in magnitude and rotate clockwise (counterclockwise) in the northern (southern) hemisphere to couple friction-driven surface winds with near-frictionless winds aloft (Stull, 1988). Further, when the surface stress decreases following radiative fluxes, inertia causes the wind to accelerate and the Coriolis force turns the wind vector clockwise (northern
hemisphere) in time (Stull, 1988). The opposite case occurs with increasing surface stresses.

The combined effect of speed and direction shear on turbine performance displayed a linear threshold (given in Eq. (5)) that separates under- and overperformance at this wind farm (Figure 14). Several models have shown that power law exponents between 0 and 0.33 result in lower available power over the whole rotor area (e.g. Antoniou et al., 2009; Bardal et al., 2015). Also, as the wind vector turns with height the magnitude of the projected velocity decreases following a cosine function, causing a reduction in available power. Here, we found slight overperformance for wind shear combinations below the $\alpha/\beta$
threshold and $0 < \alpha < 0.33$, suggesting turbine blades' efficiency increased for these speed and direction shear ranges.

Large wind veering combined with small speed shear resulted in wind turbine underperformance (Figure 14). In contrast, overperformance occurred for large speed shear and small changes in wind direction with height. Observations exceeding the $\alpha/\beta$ threshold suggest that large direction shear undermined turbine operation as mean normalized performance remained
below 0.96. Cases below the threshold demonstrated some underperformance (0.89 for $0.3 < \alpha < 0.4$, and direction shear between -0.1 deg m$^{-1}$ and 0 deg m$^{-1}$) compared to mean operating conditions, however, mean 10-min normalized power production remained above 1 and almost three out of five observations presented overperformance. Turbine simulations using a linear wind speed and direction change across the rotor layer by Walter et al. (2009) displayed similar results, showing small power gains for little direction shear and large speed shear. However, they found the greatest power depletion for large speed
shear ($\alpha = 0.35$) and counterclockwise direction shear (-0.472 deg m$^{-1}$), whereas our observations revealed the most detrimental conditions to be at very large veering values. Dissimilar results may be due to the scarce number of backing cases that were recorded throughout the campaign. In addition, the combined effect of speed and direction shear in this site proved to be a major factor affecting turbine performance as 54% of partial load power production took place during above-threshold atmospheric conditions.

For a given value of directional shear, as quantified in 0.1 deg m$^{-1}$ intervals, increasing the speed shear boosted the turbine performance (Figure 14). Normalized performance revealed a positive trend for each 0.1 deg m$^{-1}$ direction shear bin as the rate of change of wind speed with height grew in magnitude. Hunter at al. (2001) reports similar results, whereby a growing magnitude of the wind vector caused a positive change in turbine power production for most wind speed regimes below rated speed. Simulations by Rareshide et al. (2009) indicate a power reduction for power law exponents between 0 and around 0.5,
however observational data in their study coincides with our results for large veering and backing cases. Bardal et al. (2015), though, encountered the opposite for wind speeds in the middle of the partial load regime, possibly caused by the development of an internal boundary layer due to roughness changes at their test site owed to land/sea interfaces. Simulations by Wagner et al. (2010) displayed similar results as Bardal et al. (2015): they found a decrease in turbine power production for increasing speed shear and wind speeds above 5 m s$^{-1}$. Dissimilar theoretical and experimental results have been previously reported

(Hunter et al., 2001). Further, though our findings indicate overperformance for rising power law exponents, the rate of increase in normalized power was smaller for large speed shear, possibly suggesting a decrease in turbine efficiency akin to simulations by Antoniou et al. (2009). In addition to differences in boundary-layer structure (like the internal boundary layer of Bardal et al. (2015)), we must point out that differences in types of turbines, turbine blade design, and turbine control algorithms may

influence these results.

       For a given value of speed shear, as quantified in 0.1 power law exponent intervals, increasing the directional shear resulted in turbine power depletion at this wind farm (Figure 14). Normalized performance revealed a negative trend (around -0.04 per increase in 0.1 deg m$^{-1}$ of direction shear) for all 0.1-power law exponent bins as the change of wind direction with height grew in magnitude. Likewise, blade-element modelling using FAST evidenced decreasing turbine performance for increasing

wind veer for all speed shear exponents between 0 and 0.6 (Walter et al., 2009). Conversely, Rareshide et al. (2009), looking exclusively at 8 m s$^{-1}$ wind speeds, only reported underperformance for speed shear exponents around 0.2 and veering near 0.25 deg m$^{-1}$. Our results suggest more notable underperformance to occur for larger direction shear values, which were not considered in their study. Contrasting results between Walter et al. (2009) and Rareshide et al. (2009), and our findings for backing cases between $0 < \alpha < 0.6$ may be due to much less frequent and smaller numerical values compared to wind veering

in our dataset. Simulations by Wagner et al. (2010) depict similar results, yet slight underperformance also occurred for veering values above 0.2 deg m$^{-1}$ and wind speeds above 8 m s$^{-1}$.

       Small wind backing was found to have similar effects as small wind veering. The change in energy flux through the rotor disc and turbine blades' efficiency appeared to be minor for these low direction shear conditions. Our dataset only evidenced statistically distinct power production between veering and backing for two speed shear ranges, suggesting the power

asymmetries found by Walter et al. (2009) and Wagner et al. (2010) did not occur at these low shear conditions. Moreover, the small mean backing numerical values for these speed shear ranges indicate additional forcing mechanisms were in place for these underperformance observations. Not enough large backing observations were recorded to compare turbine performance against large veering atmospheric conditions.

       In distinguishing the effects of high- and low-direction shear using the $\alpha/\beta$ threshold over 0.1-$\alpha$ ranges, large wind veer

reduced power output by more than 10% compared to below-threshold scenarios for wind speeds in the middle of the power curve (Figure 15). The larger proportionality between shear parameters found at lower wind speeds (Figure 13) may have augmented the effect of shear on turbine performance. These findings support those found by Bardal et al. (2015), where wind veer larger than 5 deg over a 100 m rotor layer (0.05 deg m$^{-1}$) was found to have its major effects in the middle of the power curve, still, the affected wind speeds differ. As stated earlier, these incongruencies may be caused by dissimilar boundary layer

structures given that most of the analyzed winds came from offshore in their case.

       When considering power law exponents between 0.2 and 0.3, we found direction shear to exert a larger impact on power production in the middle of the partial load regime than near cut-in or rated speeds (Figure 15a). Most observations within this speed shear range took place between 6.5 and 8 m s$^{-1}$ (Figure 12), corresponding to the most affected turbine performance-speed regimes. On the other hand, highly stratified atmospheric conditions, characterized by large speed shear ($0.4 < \alpha < 0.5$),

evidenced statistically distinct power differences for larger wind speeds (Figure 15b). Likewise, most observations for this speed shear range corresponded to near-rated wind speeds. We expected mean normalized performance for above-threshold scenarios during highly stratified atmospheric conditions ($0.4 < \alpha < 0.5$) to be smaller compared to power law exponents between 0.2 and 0.3, nonetheless observational data proved opposite. Larger directional wind shear thresholds for the former

cases suggested analogous underperformance, however, the mechanical turbulence that usually accompanies large speed shear may have influenced turbine operation as well. These results prove direction shear to be an important factor that influences turbine operation. Moreover, more than 35% of observations for moderate speed shear values ($0.2 - 0.5$) and their correspondingly statistically affected wind speed regimes of each 0.1-power law exponent bin occurred for above-threshold shear conditions.

Focusing on a period of rapidly increasing electricity demand (0600 to 0900 local time) exposed the fact that directional shear's detrimental effects preferentially occurred during this time. Mean direction and speed shear were 0.196 deg m$^{-1}$ and 0.37, respectively. Mean normalized power reductions were larger for this time period ($\sim$20%) compared to whole-day results ($\sim$10%) for statistically distinct wind speeds between 4.5 and 10.5 m s$^{-1}$ and speed shear between 0.2 and 0.5 (normalized power calculated for each 0.1-$\alpha$ bin). Further, around 22% of observations for this time presented speed shear exponents

between 0.3 and 0.4, which evidenced mean normalized power reductions close to 21% for six out of the eight 0.5-m s$^{-1}$ wind speed regimes between 5 and 9 m s$^{-1}$ (Figure 16). Not only did large wind shear occurred often during this high-demand period of the day, but it also undermined power production at this time.

The substantial power reductions and number of cases affected by the change of wind direction with height in this wind farm make directional wind shear effects critical to consider in wind resource assessment, grid integration studies, and wind

turbine control algorithm design. Large veering values affected turbine performance for small and large speed shear, suggesting that aerodynamic efficiency reductions dominate the increase in energy flux over the rotor disc caused by increasing speed shear values. In addition, the fact that large directional shear undermined power production here also provides an explanation for how turbine operation was undermined for stable atmospheric conditions in the Vanderwende and Lundquist (2012) study. Turbine overperformance for stratified channeled flow conditions in St. Martin et al. (2016) and Wharton and Lundquist

(2012b) studies was likely augmented by low direction shear due to channeled flow in those regions.

The present work has provided insight into the impact of wind veer on clockwise-rotating wind turbines' performance for different wind shear conditions. Recent simulations suggest that the direction of turbine rotation interacts with wind veer to affect wake structures (Englberger et al., 2019). To understand the impact of the rotational direction of a wind turbine on performance, however, future field studies and simulations should incorporate counterclockwise-rotating wind turbines.

Further work regarding directional wind shear in offshore locations should also be pursued. A preliminary wind resource assessment on the coast of Massachusetts by Bodini et al. (2019) demonstrated large changes in wind direction with height. Average values of 0.1 deg m$^{-1}$ for summertime, and 0.05 deg m$^{-1}$ for wintertime (Bodini et al., 2019) approach the threshold at which we found significant power reductions for speed shear exponents between 0.2 and 0.3. Further, they also found the

summer to have low turbulence dissipation rates, thus long-propagating skewed wakes may impact power production and loads on downwind turbines.

**Acknowledgements**

The CWEX project was supported by the National Science Foundation under the State of Iowa EPSCoR grant 1101284. The role of the University of Colorado Boulder in CWEX-13 was supported by the National Renewable Energy Laboratory. The authors thank NextEra Energy for providing the wind turbine power data. This work was authored [in part] by the National Renewable Energy Laboratory, operated by Alliance for Sustainable Energy, LLC, for the U.S. Department of Energy (DOE) under Contract No. DE-AC36-08GO28308. Funding provided by the U.S. Department of Energy Office of Energy Efficiency and Renewable Energy Wind Energy Technologies Office. The views expressed in the article do not necessarily represent the views of the DOE or the U.S. Government. The U.S. Government retains and the publisher, by accepting the article for publication, acknowledges that the U.S. Government retains a nonexclusive, paid-up, irrevocable, worldwide license to publish or reproduce the published form of this work, or allow others to do so, for U.S. Government purposes.

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
