# Peer review of "The effect of wind direction shear on turbine performance in a wind farm in central Iowa"

_Wind Energy Science, 2019_

## Referee Comment (RC1) · Anonymous Referee #1 · 24 Jun 2019

**Review of WES-2019 22:** 'The effect of wind direction shear on turbine performance in a wind farm in central Iowa' by Miguel Sanchez Gomez and Julie K. Lundquist

The authors present a statistical analysis of wind power of four wind turbines and wind direction shear, measured using a nearby Lidar. They discriminate between clockwise and counter-clockwise wind direction shear and show a correlation between under performance of the wind turbine and mean directional wind shear. This under performance is primarily visible for lower wind speeds. In general, the paper is very well written. Introduction, choice of data and statistical analysis are clear. The figures are suitable to present the data. My comments are therefore very minor asking for some clarification and some improvements of text and figures. I recommend that the paper is accepted after minor revisions.

**General comments**

The fact that turbine performance decreases by more than 15% for strong wind shear conditions is very relevant; I can clearly see that. I found the fact that is is mostly high veering that is related to a drop in normalized power very interesting. However, I was missing any attempt for a physical explanation on why high veering has such a strong effect whereas high backing does not show the same signal. It seems that you are too defensive here, and a little bit of speculation could be appropriate in the discussion.

Furthermore, I wonder if the unit deg m$^{-1}$ is indeed the right one in this context. In my opinion, it depicts a certain generality that cannot be drawn from your dataset, given that you looked only at four wind turbines with the same rotor diameter. Given that directional shear is not linear, this may be relevant. Multiplying by 80 and using deg D$^{-1}$ may help to communicate this important limitation.

Third, I found that the directional wind shear is a function of height was overlooked in the analysis. This fact is clearly stated in the beginning, but is ignored later on and only average directional shear is analysed. I wonder especially, what the effect of misalignment (with respect to the ideal 90 degree angle) would be if this misalignment is primarily below or above hub height. Given that wind speed generally increases with height, this seems important. You probably lack data on rotor orientation, but could you expand on this point in the discussion nonetheless? Splitting the analysis to the levels 40-80 and 80-120 may provide some insight.

Finally, the rotation direction of the rotor is one asymmetry that may be of relevance given that you found such a distinct difference between veering and backing. Any thoughts?

**Specific comments**

1. *Section 2.4:* Do you really need a section of its own for two sentences? Furthermore you mention that you average over 10 minutes in line 16, page 7 as well.

2. *Page 15, line 13:* Mentioning Turbine A-D comes a little bit out of the blue at this point because this is the very first time the text and the only other reference is Figure

1. You should refer to Figure 1 in this context.

3. *Figure 12:* I found it confusing that there are four entries in the legend, but only three lines. You write in the text that bins with less than 30 data points are discarded, but I recommend to remove the entry for high backing. Furthermore, it is rather strange that the error bars for the low veering case are not bold and black. It is a minor issue, but your readers should not play a guessing game. I have similar issues with Figs. 11 and 13, having four lines, but only 2 legend entries (specifically referring to the median).

4. *Figure 2:* The shading in this plot style leads to three colors in the graph, while there are only two in the legend. It is only a very,very minor remark, but wouldn't a simple line plot do a better job in communicating data availability?

**Typos**

1. *Page 8, Equation 1:* Full stop instead of komma after the equation.

2. *Page 15, line 7:* 'was on average' instead of 'was in average'

3. References For Rajewski et. al 2016 and Muñoz-Esparza et.al. 2017, it seems that there are some personal comments that have slipped from the .bib file into the references. The all-caps text is not part of the actual reference.

4. *Figures 11,12,13:* You are using "Wind Speed"in the x-label here, but "Wind speed"everywhere else.

---

## Referee Comment (RC2) · Rozenn Wagner (Referee) · 17 Jul 2019

General comment

Interesting topic. There is definitely a need for studying and publishing on the effect of veer on wind turbine power performance. However the usual challenge with experimental data is how to identify whether the effect on the performance is due to veer or other wind conditions, especially speed shear which is very much correlated with veer. This is my main criticism to the paper: all the observed performance variations are attributed to the veer effect without considering the speed shear during the measurement campaign. This could totally invalidate the results. A description of the shear during those measurements and an estimate of its expected effect on the turbine performance need to be included in the paper.

Apart from that, the paper is rather well structured. References are made to the relevant existing study on the matter. The analysis and interpretation of the results need to be clarified in several places but this probably only a matter of sharpening the language (see detailed comments).

Detailed comments

P3- l.9: could you clarify this sentence or paragraph? Do you mea in Vanderwende, it was not complex terrain contrary to the two other studies or it was complex terrain in all of them – then why does complex terrain prevents the occurrence of changing wind direction wind height in the two first cases and not in the other?

P3-l.22,24 and 25: wrong cross ref: "section 0"

P4, l3-4 The data have also been used…LES models" – delete this sentence. It is not relevant in the context of this study.

P4- Figure 1: where is this sub-region located in relation to the overall 200 turbines wind farm? Are there any turbine or any obstacles to the south of that region? How far?

P5, l8: why have you used 2-min averages and not 10 min?

P5, fug.2: how were the availability calculated for the lidar and turbine? Availability of 2 min data over one day? At what height for the lidar? For which turbine? Four turbine were mentioned in figure 1? Were the data for wind speed below 2.5m/s also excluded from the lidar data for availability quantification?

P7, fig. 4: those data are from which of the 4 turbines highlighted in Fig 1?

P7, section 2.4

1. This is a bit confusing since it is stated in page 5 that the study was based on 2min averages. Could you please clarify?
2. What is meant by "average over different time periods": the 10 min period were named after the beginning of the period for the turbine and the end of the period for the lidars(or vice versa) and it could be fixed with a  10 min shift or the clocks were not synchronized during the campaign and you had to manually find the beginning of the 10 min period to be averaged?
3. How have you checked the time synchronization between the turbine and lidar signals?
4. This section about time averaging and time ynchronisation should be moved before the "wind turbine" section since the time averaging and synchronization needed to be performed before producing the plots in figure 4.

P8, l3-4 (and figure 5):

1. How come the measure power is larger than for wind speed above 12m/s? this does not match with the blue dots in Figure 4a).
2. For what range of wind speed shear, direction shear and TI was the power curve provided by the manufacturer? Could that explain the discrepancy between 5 and 8m/s?

P8, l7 to 11. I do not quite understand the point of the exercise with the Pearson correlation coefficient.

1. What is the purpose of this comparison? Did you have some doubt about the lidar measurements accuracy? Has it been compared to another wind speed measurement (e.g. mast mounted anemometer or other lidar) before or after the campaign? Or is this to show somehow that there is a good correlation between the lidar wind speed and the turbine power, although it was placed rather far from the turbine(s)?
2. The nacelle anemometer measurements are expected to be disturbed by the rotor and the nacelle. The manufacturer power curve is provided with the free hub height wind speed on the x-axis. In order to get a comparable measured power curve using the nacelle wind speed, it must be corrected with a Nacelle Transfer Function (NTF)? Was this applied? If so, where did the NTF come from? If the nacelle wind speed was not corrected, it is to be expected that there is not be a good agreement with the manufacturer power curve.

p.9: "Directional shear": this section is hanging (missing numbering)

p9, lines 6 to 17: this should be in the introduction.

P11, figure 7: there are average and median over the whole measurement period? Could you include a standard deviation or 25th, 75th percentile in the figure as well?

P12, l.9: how does the LDWS and HDWS events correlate to wind speed shear and turbulence intensity?

 Figure 10 to 13:

1. are those results for Turbine D?
2. strong veer is expected to be correlated with strong speed shear – which is expected to have a larg impact on the power performance. How do you know those results are not just due to the wind speed shear?

p18, l.9 to p19, l3.The points you are trying to make with comparison to other studies need to be clarified. The reasons why you finds them in contradiction with your own results are not clear.

P19, l15-17. This is often the challenge. But is no because the dataset is too small to bin them according to spped shear and direction shear (with statistical significance in al bins) that the effect of speed shear can be ignored. The speed shear or wind speed profile for this specific dataset has to be carefully understood and described in the paper before drawing any conclusions on the effect of veer on the turbine power performance.

P19, l21 to 33. I think it is outstretching to expand the discussion to skewed wake effect on power performance here (how do you make the different between the underperformance due to "standard "wake and skewed wake?) and is out of scope.

Looking forward to reading you again. And thanks for giving me the chance to get back to this topic after many years.

Best regards,

Rozenn

---

## Author Comment (AC1) · 29 Aug 2019

**Response to reviewer 1**

Dear Anonymous Reviewer 1,

We appreciate your thoughtful feedback. It helped us to improve the manuscript. Below we comment on your suggestions in detail.
All reviewer comments appear in *italic text* below, while authors' responses appear in blue text. Line numbers referenced in the authors' responses refer to the revised document.

*The authors present a statistical analysis of wind power of four wind turbines and wind direction shear, measured using a nearby Lidar. They discriminate between clockwise and counter-clockwise wind direction shear and show a correlation between under performance of the wind turbine and mean directional wind shear. This under performance is primarily visible for lower wind speeds. In general, the paper is very well written. Introduction, choice of data and statistical analysis are clear. The figures are suitable to present the data. My comments are therefore very minor asking for some clarification and some improvements of text and figures. I recommend that the paper is accepted after minor revisions.*

**General comments**

*The fact that turbine performance decreases by more than 15% for strong wind shear conditions is very relevant; I can clearly see that. I found the fact that is mostly high veering that is related to a drop in normalized power very interesting. However, I was missing any attempt for a physical explanation on why high veering has such a strong effect whereas high backing does not show the same signal. It seems that you are too defensive here, and a little bit of speculation could be appropriate in the discussion.*

Thank you for giving relevancy to our findings. We agree that we did not provide a physical explanation on why high backing does not show a drop in normalized power. To provide a robust and defensible explanation, we would have liked to have a greater number of strong backing cases so that we could develop a rigorous hypothesis. Unfortunately, as we pointed out, an insufficient number of large backing cases occurred so we could not conclusively draw any conclusions regarding its effect on turbine performance. In the updated version of our manuscript, we examine in more detail the effects of small veering and small backing on normalized performance and found similar results for each scenario. We speculate that the variations in available power through the rotor disc and in turbine blades' efficiency are not significant for these subtle wind direction changes (see p. 21, lines 30 – 34; p. 22, lines 1 – 2).

*Furthermore, I wonder if the unit deg m$^{-1}$ is indeed the right one in this context. In my opinion, it depicts a certain generality that cannot be drawn from your dataset, given that you looked only at four wind turbines with the same rotor diameter. Given that directional shear is not linear, this may be relevant. Multiplying by 80 and using deg D$^{-1}$ may help to communicate this important limitation.*

We carefully considered this thoughtful suggestion, but literature regarding the effects of wind direction shear on turbine performance (e.g. Rareshide et al., 2009; Wagner et al., 2010; Walter et al., 2009) and characterization of the lower boundary layer for wind energy applications (e.g. Bodini et al., 2017) generally employs this unit. Further, comparisons made with other studies are easier to understand when maintaining the same units. Therefore, we deem appropriate using deg m$^{-1}$ for characterizing shear in wind direction. We also consulted with colleagues using veer in their wind turbine control algorithms, and they specifically requested deg m$^{-1}$.

*Third, I found that the directional wind shear is a function of height was overlooked in the analysis. This fact is clearly stated in the beginning but is ignored later on and only average directional shear is analysed. I wonder especially, what the effect of misalignment (with respect to the ideal 90 degree angle) would be if this misalignment is primarily below or above hub height. Given that wind speed generally increases with height, this seems important. You probably lack data on rotor orientation, but could you expand on this point in the discussion nonetheless? Splitting the analysis to the levels 40-80 and 80-120 may provide some insight.*

Thank you for your comment. We considered your suggestion, but as is stated at the beginning of the manuscript, the main objective of this paper was to determine if direction wind shear across the whole rotor layer affects turbine performance to see if this phenomenon might have played an important role in conflicting results regarding the effects of atmospheric stability on turbine operation. If this overall veer did not show interesting results, then breaking it up into 40-80m and 80-120m segments likely would not have done so either. Hopefully our manuscript can motivate more detailed investigations using observations or simulations.

*Finally, the rotation direction of the rotor is one asymmetry that may be of relevance given that you found such a distinct difference between veering and backing. Any thoughts?*

Thank you for highlighting this as it most certainly plays an important role for large veering and backing scenarios. However, as stated above, we had insufficient large-backing cases to draw conclusions and make a more detailed comparison with turbine performance taking place during large-veering atmospheric conditions. Moreover, we did not expand on this topic as we found small veering and small backing to have similar effects on turbine operation (see p. 17, lines 9 – 18; p. 21, lines 30 – 34; p. 22, lines 1 – 2). The reviewer may be interested to see an investigation of the interactions of veering with wind turbine rotation direction and the effects on turbine wake structure, Englberger et al. (2019), at https://www.wind-energ-sci-discuss.net/wes-2019-45/.

**Specific comments**

*1. Section 2.4: Do you really need a section of its own for two sentences? Furthermore, you mention that you average over 10 minutes in line 16, page 7 as well.*

We agree that we did not present information in the best possible way. We have now clarified information in this subsection and expanded it. One of these expansions was to accommodate another reviewer's request for information on time synchronization between turbine and lidar data. Also, we consider it important to explain that lidar data is logged every 2 minutes (2-min averages), and turbine data is logged every 10 minutes (10-min averages) by their data-acquisition systems. Therefore, we match turbine-measured power operation to lidar-measured atmospheric conditions by averaging five 2-min lidar recordings. The text now reads as follows: "Turbine- and lidar-recorded data are averaged over different time intervals by their respective data-acquisition systems (2-min for lidar, and 10-min for turbine). Matching turbine performance with atmospheric conditions was performed by averaging 2-min lidar measurements for the corresponding 10-min turbine power production period. For example, turbine data for July 04, 2013 from 0500 to 0510 LT is matched with the average of five 2-min lidar data measurements corresponding to the same date and time period".

*2. Page 15, line 13: Mentioning Turbine A-D comes a little bit out of the blue at this point because this is the very first time the text and the only other reference is Figure 1. You should refer to Figure 1 in this context.*

Thank you for pointing this out, we did so in our updated version of our manuscript.

*3. Figure 12: I found it confusing that there are four entries in the legend, but only three lines. You write in the text that bins with less than 30 data points are discarded, but I recommend removing the entry for high backing. Furthermore, it is rather strange that the error bars for the low veering case are not bold and black. It is a minor issue, but your readers should not play a guessing game. I have similar issues with Figs. 11 and 13, having four lines, but only 2 legend entries (specifically referring to the median).*

Thank you for your comment. We agree that figures should be as clear as possible for the readers. We have included corresponding legends to all our new figures in the updated version of our manuscript.

*4. Figure 2: The shading in this plot style leads to three colors in the graph, while there are only two in the legend. It is only a very, very minor remark, but wouldn't a simple line plot do a better job in communicating data availability?*

Thank you for this suggestion. We considered a line plot for the updated version of our manuscript but preferred a bar graph. However, we have now separated data availability for turbines and lidar observations to avoid overlapping colors.

**Typos**

1. *Page 8, Equation 1: Full stop instead of comma after the equation.*

   We appreciate this comment and updated the manuscript accordingly.

2. *Page 15, line 7: 'was on average' instead of 'was in average'*

   We appreciate this comment and updated the manuscript accordingly.

3. *References For Rajewski et. al 2016 and Muñoz-Esparza et.al. 2017, it seems that there are some personal comments that have slipped from the .bib file into the references. The all-caps text is not part of the actual reference.*

   We appreciate this comment and updated the manuscript accordingly.

4. *Figures 11,12,13: You are using "Wind Speed in the x-label here, but "Wind speed everywhere else.*

   Thank you for catching this. We have updated axes labels in every figure and only left the first word of each label with an initial capital letter.

---

## Author Comment (AC2) · 29 Aug 2019

**Response to reviewer 1: Rozenn Wagner**

Dear Dr. Wagner,

We highly appreciate your feedback. It helped us to improve the manuscript and strengthen our findings. You highlighted some crucial points that were overlooked in our initial manuscript.

All reviewer comments appear in *italic text* below, while authors' responses appear in blue text. Line numbers referenced in the authors' responses refer to the revised document.

*General comment*

*Interesting topic. There is definitely a need for studying and publishing on the effect of veer on wind turbine power performance. However the usual challenge with experimental data is how to identify whether the effect on the performance is due to veer or other wind conditions, especially speed shear which is very much correlated with veer. This is my main criticism to the paper: all the observed performance variations are attributed to the veer effect without considering the speed shear during the measurement campaign. This could totally invalidate the results. A description of the shear during those measurements and an estimate of its expected effect on the turbine performance need to be included in the paper.*

*Apart from that, the paper is rather well structured. References are made to the relevant existing study on the matter. The analysis and interpretation of the results need to be clarified in several places but this probably only a matter of sharpening the language (see detailed comments).*

Thank you for outlining the importance of studying the effect of direction wind shear on turbine performance. We agree that we did not cover the effect of speed shear on turbine operation in the first version, attributing performance variations solely to wind veer. We expanded the Results section to include speed shear characterization (subsection 4.1) following the same methodology as with direction shear. We also included the effects of speed shear on turbine performance (subsection 4.2) by segregating turbine operation according to speed and direction wind shear atmospheric conditions. Later on, in section 5, we further discuss our measurements in contrast to the results and findings of other studies and simulations. We feel these changes have improved the insights available from this dataset.

*Detailed comments*

*P3- l.9: could you clarify this sentence or paragraph? Do you mean in Vanderwende, it was not complex terrain contrary to the two other studies, or it was complex terrain in all of them – then why does complex terrain prevents the occurrence of changing wind direction wind height in the two first cases and not in the other?*

Thank you for pointing out we needed to be more specific here – this apparent contradiction is what drove the development of this paper. In Vanderwende and Lundquist, complex terrain was situated nearby but did not channel the flow into the wind turbines. A clarification has been added to the manuscript. The manuscript reads now as follow: "The Wharton and Lundquist (2012a,b) wind turbines, and St. Martin et al. (2016) testing site were surrounded by complex terrain that steered a channeled flow into the turbines and prevented the development of directional wind shear during stable conditions. In contrast, at the Vanderwende and Lundquist (2012) location, complex terrain did not prevent the occurrence of a changing wind direction with height."

*P3-l.22,24 and 25: wrong cross ref: "section 0"*

We appreciate this comment and updated the manuscript accordingly.

*P4, l3-4 The data have also been used...LES models" – delete this sentence. It is not relevant in the context of this study.*

We included this information to give context of the wide variety of studies that stemmed from the CWEX-13 campaign. We decided to keep the reference to this work but moved this sentence further up in the paragraph near references to low-level jet and wake variability studies.

*P4- Figure 1: where is this sub-region located in relation to the overall 200 turbines wind farm? Are there any turbine or any obstacles to the south of that region? How far?*

Thank you for asking about this issue as any upwind obstacles would have affected turbine inflow conditions. We changed the schematic diagram of the region of interest in the CWEX-13 campaign and in this study to show its location in relation to the overall 200 turbines wind farm and to show topography near turbines (refer to Figure 1 in the updated manuscript). No obstacles are located to the south of the region of interest.

*P5, l8: why have you used 2-min averages and not 10 min?*

The lidar data was recorded as 2-min averages for other purposes, to aid in meteorological studies of cold front passages and other transient phenomena. For the present work, we simply average 2-min data from the lidar over their respective 10-min turbine operating periods.

*P5, fig.2: how were the availability calculated for the lidar and turbine? Availability of 2 min data over one day? At what height for the lidar? For which turbine? Four turbines were mentioned in figure 1? Were the data for wind speed below 2.5m/s also excluded from the lidar data for availability quantification?*

We appreciate this comment and the opportunity to clarify the method for calculating data availability, expanding subsection 2.2 and extending Figure 2. Lidar data availability corresponds to lidar-operating time throughout the whole day. It was calculated as availability of 2-min observations of 80-m wind speeds over the whole day (24 hours). Turbine data availability was calculated as availability of power observations for nacelle-measured wind speed above 2.5 m s$^{-1}$. Turbine data availability showed in Figure 2 corresponds to the mean data availability of the four turbines. Subsection 2.2 has been updated to include this clarification, and data availability in Figure 2 has been divided to show lidar data availability scaled in hours, and turbine data availability in percentage of power measurements recorded for wind speeds above 2.5 m s$^{-1}$.

*P7, fig. 4: those data are from which of the 4 turbines highlighted in Fig 1?*

We have added clarification in Figure 4 caption stating that data corresponds to power and blade pitch angle measurements of the four analyzed wind turbines.

*P7, section 2.4*

1. *This is a bit confusing since it is stated in page 5 that the study was based on 2min averages. Could you please clarify?*

We agree that it seems confusing, so we explained that to determine atmospheric conditions for each 10-min power recordings we averaged the 2-min lidar data over the corresponding power production period and we added a short example. The text now reads as follows: "Turbine- and lidar-recorded data are averaged over different time intervals by their respective data-acquisition systems (2-min for lidar, and 10-min for turbine). Matching turbine performance with atmospheric conditions was performed by averaging 2-min lidar measurements for the corresponding 10-min turbine power production period. For example, turbine data for July 04, 2013 from 0500 to 0510 LT is matched with the average of five 2-min lidar data measurements corresponding to the same date and time period".

2. *What is meant by "average over different time periods": the 10 min period were named after the beginning of the period for the turbine and the end of the period for the lidars(or vice versa) and it could be fixed with a 10 min shift or the clocks were not synchronized during the campaign and you had to manually find the beginning of the 10 min period to be averaged?*

Thank you for this comment. We changed "time periods" for "time intervals" and included a clarifying parenthesis exemplifying the time intervals in which the lidar and SCADA averages measurements.

3. *How have you checked the time synchronization between the turbine and lidar signals?*

Yes, this is an important issue. We have added Figure 5 to demonstrate the synchronization between turbine and lidar signals using 80-m wind speeds.

4. *This section about time averaging and time synchronization should be moved before the "wind turbine" section since the time averaging and synchronization needed to be performed before producing the plots in figure 4.*

Thank you for this comment. However, plots in Figure 4 were obtained using nacelle-measured wind speed as recommended by St. Martin et al. (2016). We left this section last as it combines what is said in the previous two sections and it now includes synchronization between lidar and turbine measurements.

*P8, l3-4 (and figure 5):*

1. *How come the measure power is larger than for wind speed above 12m/s? this does not match with the blue dots in Figure 4a).*

Thank you for comparing these figures. However, measured power in Figure 4a also shows power above rated (measurements around 1.6 MW) for wind speeds between 11 and 14 m s$^{-1}$.

2. *For what range of wind speed shear, direction shear and TI was the power curve provided by the manufacturer? Could that explain the discrepancy between 5 and 8m/s?*

Thank you for noting this lack of information as these conditions have been previously found to alter a turbine's power curve. However, we were not able to obtain this information from the manufacturer. Still, we were not trying to find the cause for such discrepancies given that each of the four turbines demonstrated differing power curves for different wind speed regimes. For instance, while turbine D showed overperformance for wind speeds between 9 and 10 m s$^{-1}$, turbine A showed slight underperformance. Therefore, given the observed discrepancies between each turbine's mean power curve and the manufacturer's, we chose to use our mean power curve for normalization to compare turbine operation changes compared to their average operation throughout the campaign. With mean power curve as baseline, if we normalize turbine power using the mean turbine operating conditions throughout the campaign, then shifts from this mean may be due to speed or direction shear.

*P8, l7 to 11. I do not quite understand the point of the exercise with the Pearson correlation coefficient.*

1. *What is the purpose of this comparison? Did you have some doubt about the lidar measurements accuracy? Has it been compared to another wind speed measurement (e.g. mast mounted anemometer or other lidar) before or after the campaign? Or is this to show somehow that there is a good correlation between the lidar wind speed and the turbine power, although it was placed rather far from the turbine(s)?*

The purpose of this comparison was to determine which wind speed data (nacelle anemometer or lidar) to use for each turbine's mean power curve. We used the Pearson correlation coefficient to find the power curve that resembled the most that of the manufacturer (see p.8, lines 16 – 20 on revised manuscript). However, we acknowledge that each turbine's mean power curve is different, and so we do not use the manufacturer's power curve as a reference frame.

2. *The nacelle anemometer measurements are expected to be disturbed by the rotor and the nacelle. The manufacturer power curve is provided with the free hub height wind speed on the x-axis. In order to get a comparable measured power curve using the nacelle wind speed, it must be corrected with a Nacelle Transfer Function (NTF)? Was this applied? If so, where did the NTF come from? If the nacelle wind speed was not corrected, it is to be expected that there is not be a good agreement with the manufacturer power curve.*

Thank you for your comment. We did not correct the power curve obtained using nacelle wind speed with a Nacelle Transfer Function. However, as stated above, the purpose of comparing power curves obtained using lidar- and nacelle-measured wind speed was to find the power curve that resembled the most that of the manufacturer so we could normalize observations in relation to each turbine's mean operating conditions. We clarified this in subsection 3.1, the updated manuscript for p.8, lines 16-17 reads as follows: "Consequently, mean power curves for each turbine were utilized as a reference value for normalization to have a consistent comparison of performance among individual devices."

*p.9: "Directional shear": this section is hanging (missing numbering)*

Thank you for catching this, we have added numbering for this subsection.

*p9, lines 6 to 17: this should be in the introduction.*

Thank you for this suggestion. We added the main idea of this paragraph to the introduction (p.2, lines 22 – 23).

*P11, figure 7: there are average and median over the whole measurement period? Could you include a standard deviation or 25ᵗʰ, 75ᵗʰ percentile in the figure as well?*

Thank you for this comment. We agree that it is very important to include the spread of the data in every plot, and so we tried this. However, adding this information obscures the main idea of this figure given that we have also added the evolution of speed shear into the same plot. Further, the purpose of these plots is to show the general evolution of shear, and correspondence between both shear parameters with time of day. For turbine performance, though, we included standard deviations.

*P12, l.9: how does the LDWS and HDWS events correlate to wind speed shear and turbulence intensity?*

Thank you for this helpful suggestion which has motivated a lot of analysis. A description of speed shear has been included in the updated manuscript. We included speed and direction shear relationships for similar wind speeds (Figure 13) and normalized performance segregated using both shear parameters (Figure 14). However, we did not have enough data to further segregate power production according to turbulence intensity.

*Figure 10 to 13:*

1. *are those results for Turbine D?*
2. *strong veer is expected to be correlated with strong speed shear – which is expected to have a large impact on the power performance. How do you know those results are not just due to the wind speed shear?*

   Thank you for raising an interesting point. We have now included the effects of speed shear in our analysis and also considered the effect of direction shear for similar power law exponents (0.1 speed shear exponent bins). In the new Figures 15 and 16, we show the effects of direction shear for different wind speeds and 0.2 – 0.3, 0.3 – 0.4, and 0.4 – 0.5 speed shear exponents. We also included clarification in each figure's caption stating that results are for the four analyzed wind turbines.

*p18, l.9 to p19, l3. The points you are trying to make with comparison to other studies need to be clarified. The reasons why you find them in contradiction with your own results are not clear.*

Thank you for this suggestion. Though our analysis somewhat changed by taking speed shear into consideration, we tried to clarify the reasons for conflicting results with other studies. The most notable differing results correspond to the absence of underperformance for large veering observations recorded by Rareshide et al. (2009), and to asymmetries in the effects of wind veering and backing found by Wagner et al. (2010) and Walter et al. (2009).

Explanations to each diverging finding reads as follows; p.21, lines 23 – 26: "Conversely, Rareshide et al. (2009), looking exclusively at 8 m s⁻¹ wind speeds, only reported underperformance for speed shear exponents around 0.2 and veering near 0.25 deg m⁻¹. Our results suggest more notable underperformance to occur for larger direction shear values, which were not considered in their study."; p.21, lines 31 – 34 and p. 22, lines 1 – 2: "Our dataset only evidenced statistically distinct power production between veering and backing for two speed shear ranges, suggesting the power asymmetries found by Walter et al. (2009) and Wagner et al. (2010) did not occur at these low shear conditions. Moreover, the small mean backing numerical values for these speed shear ranges indicate additional forcing mechanisms were in place for these underperformance observations. Not enough large backing observations were recorded to compare turbine performance against large veering atmospheric conditions."

*P19, l15-17. This is often the challenge. But is no because the dataset is too small to bin them according to speed shear and direction shear (with statistical significance in al bins) that the effect of speed shear can be ignored. The speed shear or wind speed profile for this specific dataset has to be carefully understood and described in the paper before drawing any conclusions on the effect of veer on the turbine power performance.*

Thank you for this thoughtful suggestion. We included a complete characterization of speed shear in our manuscript (new figures 7-13) analyzing its evolution throughout the day, how it depends on wind speed, and its relationship with direction shear. We also included the combined effects of direction and speed wind shear on turbine performance (new figure 14), and isolated the effects of direction shear by considering specific power law exponent values (new figure 15).

*P19, l21 to 33. I think it is outstretching to expand the discussion to skewed wake effect on power performance here (how do you make the different between the underperformance due to "standard "wake and skewed wake?) and is out of scope.*

We agree that expanding the discussion to skewed wake effect on power performance may be out of scope, so we have deleted this paragraph.

*Looking forward to reading you again. And thanks for giving me the chance to get back to this topic after many years.*

Thank you very much for your thoughtful and insightful review – we greatly appreciate your insights. Best regards, Miguel and Julie

*Best regards, Rozenn*

---

## Author Response (AR2)

**Response to reviewer: Rozenn Wagner**

Dear Dr. Wagner,

we appreciate your feedback throughout this iterative process.

All reviewer comments appear in *italic text* below, while authors' responses appear in blue text. Line numbers referenced in the authors' responses refer to the revised document.

*Minor revisions:*

*I think you have done a very good revision job. Figure 14 is particularly interesting. Thanks for this. The paper can be published as is but I would recommend to check two small things:*

*1. Figures 4 and 6: it is strange that some of the data display power value above rated power.*

Thank you for pointing this out as it means that we did not communicated our methodology clearly enough in this section. We have added clarification and a short explanation for the measured mean power output to be above rated power on p.9, lines 18-21 that reads as follows: "Turbine power output for wind speeds outside this range (e.g. turbine overperformance between 12 and 15 m s$^{-1}$ in **Error! Reference source not found.**) is not well represented by our data given that there were insufficient observations to derive a power estimator that meets the aforementioned criteria. The remaining analysis only considers the region of the power curve where the observed mean power accurately represents the real mean turbine power, effectively restricting the subsequent analysis to winds below rated speed."

*2. p.19, line 6: did you mean "wind speed SHEAR" and "Figure 8b"?*

We have modified phrasing of this sentence to point out that wind veering and positive speed shear (wind speed increasing with height) occur more often than their respective opposite cases. Lines 2-3, pg.20 now reads as follows: "Wind shear at the test site showed more veering cases than backing cases (**Error! Reference source not found.**a) and a predominance of wind speed increasing with height (**Error! Reference source not found.**b)".

In proofreading our final version of the manuscript, we realized we did not fully explain our process, so in addition to responding to the reviewer's comment, we have added this clarification paragraph in p.16:

[revised manuscript text omitted]